# Highly Charged Ru(II) Polypyridyl Complexes as Photosensitizer Agents in Photodynamic Therapy of Epithelial Ovarian Cancer Cells

**DOI:** 10.3390/ijms232113302

**Published:** 2022-11-01

**Authors:** Luca Conti, Gina Elena Giacomazzo, Barbara Valtancoli, Mauro Perfetti, Alberto Privitera, Claudia Giorgi, Patrick Severin Sfragano, Ilaria Palchetti, Sara Pecchioli, Paola Bruni, Francesca Cencetti

**Affiliations:** 1Department of Chemistry “Ugo Schiff”, University of Florence, 50019 Sesto Fiorentino, Italy; 2Department of Experimental and Clinical Biomedical Sciences “Mario Serio”, University of Florence, 50134 Florence, Italy

**Keywords:** drug discovery, coordination complexes, phototoxicity, reactive oxygen species

## Abstract

Ovarian cancer recurrence is frequent and associated with chemoresistance, leading to extremely poor prognosis. Herein, we explored the potential anti-cancer effect of a series of highly charged Ru(II)-polypyridyl complexes as photosensitizers in photodynamic therapy (PDT), which were able to efficiently sensitize the formation of singlet oxygen upon irradiation (**Ru1**^2+^ and **Ru2**^2+^) and to produce reactive oxygen species (ROS) in their corresponding dinuclear metal complexes with the Fenton active Cu(II) ion/s ([Cu**Ru1**]^4+^ and [Cu_2_**Ru2**]^6+^). Their cytotoxic and anti-tumor effects were evaluated on human ovarian cancer A2780 cells both in the absence or presence of photoirradiation, respectively. All the compounds tested were well tolerated under dark conditions, whereas they switched to exert anti-tumor activity following photoirradiation. The specific effect was mediated by the onset of programed cell death, but only in the case of **Ru1**^2+^ and **Ru2**^2+^ was preceded by the loss of mitochondrial membrane potential soon after photoactivation and ROS production, thus supporting the occurrence of apoptosis via *type II* photochemical reactions. Thus, Ru(II)-polypyridyl-based photosensitizers represent challenging tools to be further investigated in the identification of new therapeutic approaches to overcome the innate chemoresistance to platinum derivatives of some ovarian epithelial cancers and to find innovative drugs for recurrent ovarian cancer.

## 1. Introduction

Ovarian cancer is one of the most common among gynecologic cancers and is the major cause of tumor-associated death in reproductive women [1,2]. Aggressive but asymptomatic progression frequently occurs followed by late diagnosis of advanced and metastatic stage in more than 70% of patients [3]. Surgery and chemotherapy are the major therapeutic choices, nevertheless with limited benefits, since the majority of ovarian cancer patients are initially sensitive to platinum- and taxane-based chemotherapies, which are the “golden standard” approach in ovarian cancer treatment, whereas almost half unfortunately suffer from recurrence, developing therapeutic resistance [4] in response to platinum-based chemotherapy. Therefore, innovative therapeutic strategies to overcome drug resistance are urgently needed.

Ruthenium polypyridyl complexes (RPCs) have been extensively studied and analyzed for their possibility of becoming alternative drugs in place of cisplatin [5,6,7]. Recently, their rich chemical–physical repertoire, which includes a wide range of photoluminescence characteristics, DNA binding abilities, tunable absorption properties and good singlet oxygen sensitizing features, made them ideal candidates as photosensitizer agents (PSs) in the “so-called” photodynamic therapy (PDT). Their employment in this therapeutic approach continues to attract increasing attention due to the encouraging results obtained in the treatment of a wide variety of cancers, such as lung, bladder, skin tumors [8,9,10,11], and also bacterial infections [12,13,14].

In PDT, the PS is activated through irradiation with low-energy light to sensitize the generation of highly reactive species, namely reactive oxygen species (ROS), capable of ultimately leading to cytotoxic effects. This peculiar mechanism of action guarantees a complete spatial and temporal control over drug activation and thus offers the crucial advantage of potentially lowering the severe dose-limiting side effects normally occurring with standard therapeutics.

In general, ROS can be produced through distinct pathways. According to *Type I* reactions, the deactivation of the excited PS can occur via a direct electron or proton transfer to the surrounding biological substrates, leading to radical species that further interact with molecular oxygen to form ROS, such as superoxide, peroxides and hydroxyl radicals. *Type II* mechanisms are instead based on the direct interaction between the excited PS and ground-state molecular oxygen (^3^O_2_) to produce singlet oxygen (^1^O_2_), a highly cytotoxic species that is capable of rapidly reacting with biological targets (estimated half-life < 40 ns and radius of action of the order of 20 nm, in a biological environment) [15], leading to topical oxidative damages and, ultimately, to cellular death.

We have previously reported on the potential as PS agents of two highly charged Ru(II)-polypyridyl complexes [Ru(phen)_2_**L’**]^2+^ (**Ru1**^2+^) and [Ru(phen)_2_**L’’**]^2+^ (**Ru2**^2+^), featuring the peculiar polyazamacrocyclic units **L’** and **L’’** (**L’** = 4,4′-(2,5,8,11,14-pentaaza [15])-2,2′-bipyridilophane, **L’’** = 4,4′-bis-[methylen-(1,4,7,10-tetraazacyclododecane)]-2,2′ bipyridine) and their corresponding copper(II) complexes [Cu**Ru1**]^4+^ and [Cu_2_**Ru2**]^6+^ (see Figure 1) [16,17]. Compared to the majority of RPCs used in PDT, the high number of easily protonable nitrogen groups gathered on the unique polyazamacrocyclic ligand of **Ru1**^2+^, or on the two distinct cyclen (1,4,7,10-tetraazacyclododecane) moieties of **Ru2**^2+^, confers to the resulting compounds the possibility to form highly charged species in aqueous media. This ensures excellent water solubility, a key requisite for biological application, but it also strengthens the capacity of ruthenium complexes to interact with possible biological targets, such as DNA. Importantly, the polyamine-based frameworks do not alter the good ^1^O_2_ sensitizing properties of **Ru1**^2+^ and **Ru2**^2+^, making them appealing PSs for the generation of ^1^O_2_ directly in aqueous matrices.

Furthermore, **L’** and **L’’** allow stably binding up to two Fenton-active copper(II) ion/s, leading to the generation of mixed Ru(II)/Cu(II) complexed species, namely [CuRu(phen)_2_**L’**]^4+^ ([Cu**Ru1**]^4+^) and [Cu_2_Ru(phen)_2_**L’’**]^6+^ ([Cu_2_**Ru2**]^6+^). These heteronuclear forms are able to generate other types of ROS, such as hydroxyl radicals, and thus may represent versatile tools in the research of alternative cytotoxic pathways to the singlet oxygen sensitization.

Prompted by the encouraging results previously displayed by **Ru1**^2+^ and [Cu**Ru1**]^4+^ in a human melanoma cell line [16], in this work, we explored the potential as photoresponsive anti-cancer compounds of all the **Ru1**^2+^, [Cu**Ru1**]^4+^, **Ru2**^2+^ and [Cu_2_**Ru2**]^6+^ complexes in a unique and comparative study, where, in addition to the first comparison between the in-solution properties of these compounds, their in vitro anti-tumor efficacies were evaluated on A2780 human ovarian cancer cells. Moreover, a particular emphasis was placed on the study of the molecular mechanisms responsible for the observed anti-cancer effects by investigating the occurrence of apoptotic cell death, ROS production and mitochondrial function.

Our results show that ovarian cancer cells were capable of internalizing the ruthenium complexes at 6–24 h of incubation. Moreover, under dark conditions, these compounds exhibited extremely low cellular toxicity, which was particularly evident in non-cancer cells, whereas after photosensitization, they exerted a significant anti-tumor effect.

The results provided by this study can represent an important step forward in the research of alternative therapeutic approaches to platinum-based chemotherapy by employing ruthenium-based photoresponsive compounds.

## 2. Results

### 2.1. Protonation, Metal Binding of Ruthenium Compounds and Stability of the Mixed Ruthenium/Copper Complexed Species

The acid-base properties of **Ru1**^2+^ and **Ru2**^2+^ were investigated by means of potentiometric measurements in NMe_4_Cl 0.1 M at 298 ± 0.1 K. The LogK values for the protonation constants and the corresponding distribution diagrams of the species present in solution are, respectively, reported in Appendix A. As shown, the presence of polyazamacrocycles **L’** and **L’’** confers to the corresponding ruthenium compounds the ability to bind up to five ([H_5_**Ru1**]^7+^) or six ([H_6_**Ru2**]^8+^) protons in the overall range of pH investigated (between 2.5 and 10.5). Among the different protonated forms, the di- and tetra-protonated species [H_2_**Ru1**]^4+^ and [H_3_**Ru2**]^5+^ are the most abundant around neutral pH values (Appendix A), including at the physiological pH value employed in biological experiments (vide infra). Nevertheless, for the sake of clarity, these species are simply referred to as **Ru1**^2+^ and **Ru2**^2+^ throughout the manuscript.

In addition to the ability to easily protonate in aqueous solution, the nitrogen donors gathered on the polyamine **L’** and **L’’** frameworks can also act as suitable anchoring sites to host additional metal ions. Herein, we exploited this property to afford the formation of mixed heteronuclear Ru(II)/Cu(II) complexed species to evaluate whether the presence of Fenton-active Cu(II) center/s within the polyamine pockets of ruthenium compounds may have an influence on the biological potential of such hybrid, heteronuclear systems.

Analogously to the acid-base study, the formation of Ru(II)/Cu(II) complexes in solution was followed via potentiometric measurements; the LogK values for Cu(II) complexation by **Ru1**^2+^ and **Ru2**^2+^ and the corresponding distribution diagrams are, respectively, reported in Table 1 and Appendix A. As shown in Table 1, **Ru1**^2+^ and **Ru2**^2+^ form stable mono- and dinuclear complexes with Cu(II), with LogK values of 15.34 and 27.6 for the coordination of one (**Ru1**^2+^) and two (**Ru2**^2+^) Cu(II) ions. The coordination of copper is maintained in a wide range of pH, and, at the pH of the biological tests (7.4), these complexes are mainly present in their mononuclear [Cu**Ru**1]^4+^ and binuclear [Cu_2_**Ru2**]^6+^ forms (Appendix A).

Since different cations are naturally present in the biological environment, the metal-binding properties of ruthenium compounds toward other relevant metal ions were also considered. In particular, we focused on K^+^, Na^+^, Ca^2+^, Mg^2+^ and Zn^2+^, taken as the most abundant alkaline, alkaline-earth and transition cations in the cellular and extracellular matrices. As shown in Table 1, the most stable complexes among these cations were formed by Zn(II), with LogK values of 8.90 and 20.33, respectively, for the addition of one Zn(II) to **Ru1**^2+^ and two Zn(II) ions to **Ru2**^2+^. However, these values are considerably lower (up to ca. 1.7-fold) compared to Cu(II), thus highlighting the higher stability of the Ru(II)-Cu(II) complexed species compared to the ones formed by all the other cations tested. On the other hand, the affinity toward K^+^ and Na^+^ emerged to be too weak to permit an accurate determination of the relative LogK values via potentiometric analysis (LogK < 1.5).

The higher affinity of Ru(II) compounds for Cu(II) ion/s is further underlined by the selectivity diagrams reported in Appendix A and determined as previously described [18]. As shown, the presence of Zn(II), Ca(II) and Mg(II), even in high concentrations, does not affect the formation of the Cu(II) complexes by both **Ru1**^2+^ and **Ru2**^2+^, and no Cu(II) release due to displacement by other metals takes place in the investigated range of pH. Metal decomplexation can only occur in low percentage (c.ca. 10%) at more acidic pH values, as expected, considering the protonation of polyamine residues of ruthenium compounds, which competes with metal binding in strong acidic conditions.

Therefore, taken together, these data underline the high stability of [Cu**Ru1**]^4+^ and [Cu_2_**Ru2**]^6+^ in the adopted experimental conditions, suitable for the subsequent biological studies.

Lastly, it should also be mentioned that the coordination of Cu(II) markedly affects the fluorescence emission of the “metal-free” forms of ruthenium compounds. Indeed, **Ru1**^2+^ and **Ru2**^2+^ are highly luminescent and display an almost identical emission profile with a maximum centered at around 600 nm. Conversely, the presence of one ([Cu**Ru1**]^4+^) or two ([Cu_2_**Ru2**]^6+^) Cu(II) ions within their polyamine pockets causes a tight decrease in the fluorescence emission (Appendix A), an effect that can be naturally attributed to the paramagnetic nature of Cu(II) ion/s.

### 2.2. Reactive Oxygen Species (ROS) Production by Ruthenium Compounds

A key requisite for a candidate PS for PDT relies on its capacity to effectively produce ROS upon irradiation, such as the highly oxidant singlet oxygen ^1^O_2_ species, which is produced according to *type-II*-based processes [19].

However, *type I* pathways can elicit severe damages as well [20]. Moreover, *type I* and *II* mechanisms can occur simultaneously, and recent studies underlined that radical species generated from *type I* processes can cooperate with ^1^O_2_ to amplify the resulting PDT response, even under hypoxic conditions [21,22]. Therefore, the knowledge of the accessible pathways to PS agents is of paramount importance for their application in PDT.

As previously reported, **Ru1**^2+^ and **Ru2**^2+^ possess good singlet oxygen sensitizing properties, with comparable quantum yields (φ_Δ_), respectively, of 0.29 ± 0.06 and 0.38 ± 0.08 (λ_irr_ = 400 nm, CH_3_CN air-saturated solutions) [16,17]. On the contrary, the sensitization of ^1^O_2_ becomes almost completely lost when Cu(II) is bound within the polyamine pockets of compounds. This effect can be easily rationalized with the fast deactivation of the excited states of [Cu**Ru1**]^4+^ and [Cu_2_**Ru2**]^6+^ through internal conversion, which competes with the energy transfer to molecular oxygen.

For this reason, herein, we investigated the ability of [Cu**Ru1**]^4+^ and [Cu_2_**Ru2**]^6+^ to elicit the formation of different typologies of ROS. This was performed through electron paramagnetic resonance (EPR), employing 5,5-dimethyl-1-pyrroline-N-oxide (DMPO) as the spin trap agent for free radicals (see Appendix A for further details). Figure 1 reports a background experiment collected for a solution containing only H_2_O_2_ as co-reagent and DMPO, showing essentially no signal for ROS generation (blue trace in Figure 1). However, when [Cu**Ru1**]^4+^ or [Cu_2_**Ru2**]^6+^ were added to the mixture, an EPR signal appeared (green and red traces for [Cu_2_**Ru2**]^6+^ and [Cu**Ru1**]^4+^, respectively), clearly indicating ROS production by the mixed Ru(II)/Cu(II) complexes. The spectra recorded in the presence of [Cu**Ru1**]^4+^ and [Cu_2_**Ru2**]^6+^ were strikingly similar, suggesting a comparable efficiency of the two systems in producing ROS.

The analysis of the narrow EPR signals can also provide useful information regarding the nature of the produced radical species. As shown by the black trace in Figure 1, the best simulation of the experimental spectra was obtained by considering the production of a 50:50 mixture of hydroxide (OH^●^) and perhydroxyl (HOO^●^) radicals (EPR lines referring to the two radical species are labeled with symbols in Figure 1, and a simulation of the single contributions is reported in Appendix A). In analogy to other studies [23,24], these species might be the result of Fenton/Fenton-like processes mediated by the presence of reducing agents and involving the Cu(II)/Cu(I) redox cycle/s. This would be of great relevance for the biological behavior of these compounds. In fact, the cellular environment typically contains high concentrations of common reducing agents, such as glutathione, ascorbic acid and NADH, just to cite a few, which can reduce the coordinated Cu(II) ions and therefore facilitate the occurrence of Fenton/Fenton-like pathways (e.g., Cu(I) + H_2_O_2_ → Cu(II) + OH^−^ + OH^●^) [23]. On the other hand, oxidation-reduction potential (ORP) measurements and cyclic voltammetry (CV) analysis of aqueous solutions of [Cu**Ru1**]^4+^ and [Cu_2_**Ru2**]^6+^ confirmed that the copper centers of the two heteronuclear compounds might be reduced under these conditions (see paragraph 5 of Appendix A for further details).

### 2.3. Internalization of Ru(II) Complexes

The human ovarian cancer cell line A2780 has been established from an untreated patient bearing an ovarian adenocarcinoma, and it is commonly used as a model for ovarian cancer in particular to test the anti-cancer potency and delivery of various drugs [25]. Preliminarily, the analysis of Ru(II) complexes’ uptake in cancer versus non-cancer cells showed that, after 24 h of incubation, **Ru1**^2+^, **Ru2**^2+^, [Cu**Ru1**]^4+^ and [Cu_2_**Ru2**]^6+^ complexes were finely localized in discrete areas of A2780 cells, whereas they were undetectable in C2C12 myoblasts (Appendix A), thus demonstrating that ovarian cancer cells, but not untransformed myoblasts, efficiently internalize Ru(II) complexes.

The kinetics of internalization of Ru(II) complexes in A2780 cells as well as their intracellular distribution was therefore checked, employing laser-scanning confocal microscopy, with the purpose to set the proper time of incubation before photoactivation by exploiting the intrinsic fluorescence properties of Ru(II) compounds. As shown in Figure 2A, a localized distribution of **Ru1**^2+^ and **Ru2**^2+^ complexes was barely detectable in cells after 15 min, whereas the internalization increased at 6 h of incubation, showing a plateau at 24 h. Notwithstanding, the Ru(II)-Cu(II) complexes featured a considerably less intense fluorescence emission than **Ru1**^2+^ and **Ru2**^2+^ (Appendix A). Their residual emission was sufficient to monitor their cellular uptake over time, which occurs with a kinetic profile similar to the ones of **Ru1**^2+^ and **Ru2**^2+^.

In parallel with confocal microscopy, fluorometric analysis was used to evaluate the kinetics of internalization of Ru(II) compounds. A2780 cells were treated with Ru(II) complexes at 10 μM for the indicated time of incubation. The results, shown in Figure 2B, are comparable with those obtained by confocal microscopy; the fluorescence signal at 600 nm increased from 6 to 18 h, reaching a plateau at 24 h of incubation. Notwithstanding, **Ru1**^2+^ and **Ru2**^2+^, almost equally emissive when administered at the same concentration, differed with regard to the intensity of fluorescence, being higher in the latter, thus suggesting an enhanced cellular internalization of **Ru2**^2+^.

Although the intrinsic fluorescence emission of compounds made it possible to follow their respective kinetics of internalization, it did not allow any quantitative estimation due to the different emissive properties, mainly between Cu(II)-free and Cu(II)-containing complexes. To this aim, the uptake of Ru(II) complexes in A2780 cells was also evaluated by measuring the content of ruthenium in cell lysates by ICP analysis, following 24 h incubation with a 10 µM dose of each compound (Figure 2C). As shown, **Ru1**^2+^ and **Ru2**^2+^ displayed higher internalization capacities compared to [Cu**Ru1**]^4+^ and [Cu_2_**Ru2**]^6+^, the **Ru2**^2+^ compound being one with the highest cellular uptake, in good agreement with confocal microscopy and fluorometric analysis.

Moreover, with the purpose of obtaining a hint on the possible internalization pathway of photosensitizers in A2780 cells, we performed immunofluorescence analysis using antibodies against Rab5, a crucial regulator of endocytosis, employed as a marker of early endosomes. Ru(II) complexes were differently localized compared to Rab5 (Appendix A), ruling out the possible involvement of a Rab5-dependent pathway in the internalization pathway of Ru(II) complexes.

Finally, based on these results, we chose to set the time of incubation with Ru(II) complexes before photoactivation at 24 h.

### 2.4. Effect of Ru(II) Complexes on A2780 Cell Survival after Photosensitization

The dose-dependent effect of **Ru1**^2+^, **Ru2**^2+^, [Cu**Ru1**]^4+^ and [Cu_2_**Ru2**]^6+^ on dark cytotoxicity and photoactivity was evaluated through MTT assays in A2780 cells incubated for 24 h with different concentrations of Ru(II) complexes and exposed or not to photoirradiation 48 h before being analyzed. In the photoirradiation experiments, cells were irradiated with a light-emitting diode (LED, λ_max_ = 434 nm, 30 W) by employing the experimental set-up sketched in Appendix A.

As shown in Figure 3, under dark conditions, cell survival was only slightly affected by **Ru1**^2+^ and **Ru2**^2+^, at least up to a 1 μM dose of compounds. Then, beyond this value, cell viability underwent an approx. 35% decrease. An even lower cytotoxicity was displayed by the [Cu**Ru1**]^4+^ and [Cu_2_**Ru2**]^6+^ complexes, being almost ineffective within the 0–10 μM range of concentration tested. In this respect, it can be tentatively speculated that the inferior activity of the mixed Ru(II)-Cu(II) complexed species would be associated with their lower capacity to be internalized by A2780 cells, as suggested by the internalization experiments described above.

Strikingly, the irradiation of complexes triggered a significant anti-cancer effect. Marked differences between the activities in dark and upon irradiation were indeed displayed by the good singlet oxygen sensitizers **Ru1**^2+^ and **Ru2**^2+^, starting from 100 nM.

Among the mixed heteronuclear complexes, [Cu**Ru1**]^4+^ exhibited a sharper increase in phototoxicity when dosed at 10 μM, resulting in an approximatively 75% decrease of cell viability. Given the scarce ability of [Cu**Ru1**]^4+^ and [Cu_2_**Ru2**]^6+^ to sensitize the formation of singlet oxygen, the anti-survival data suggest that alternative light-mediated pathways are made accessible by these systems in the cellular environment. Considering the redox activity of heteronuclear compounds (see Section 2.2), and in analogy to our previous study [16], a synergetic action between the Fenton-active copper center/s and light to generate harmful ROS species can be envisaged.

Interestingly, the effect of [Cu_2_**Ru2**]^6+^ was considerably less pronounced compared to the one of [Cu**Ru1**]^4+^. This would suggest that, in addition to the similar cellular internalization and modes of activation of these latter two compounds, other less predictable features (such as differences in the chemical structures, chemical–physical properties, etc.) may play a role in the biological response of this typology of compounds.

Although PDT is a minimally invasive procedure, which effectively kills tumor cells, photosensitizers may have cytotoxic effects on normal cells. To study the effect of Ru(II) complexes on non-cancer cells, we performed MTT analysis in C2C12 myoblasts under dark and photoactivation conditions to assess Ru(II) complex cytotoxicity and photoactivity, respectively. As reported in Appendix A, Ru(II) complexes show negligible cytotoxicity and a minor photoactivity in myoblasts compared to A2780 cells, in agreement with the negligible internalization capacity of the PSs in this non-cancer cell model (Appendix A).

### 2.5. Effect of Ru(II) Complexes on Apoptosis of A2780 Cells after Photoactivation

To dissect the molecular mechanism responsible for the selective anti-survival effect exerted by Ru(II) complexes, the involvement of apoptosis was studied by using different approaches in A2780 cells. As shown in Figure 4A, the treatment with each photosensitizer was completely ineffective on caspase 3 activity under dark conditions, whereas light irradiation of Ru(II) complexes after 24 h incubation caused a significant and potent activation of the pro-apoptotic enzyme.

Moreover, the involvement of caspase 3 in the mechanisms of action of Ru(II) photosensitizers was further investigated in dark conditions and after photoactivation employing Western blot analysis by measuring the proteolytic cleavage of the enzyme (Figure 4B). Although each Ru(II) complex was not able to induce caspase 3 cleavage under dark conditions, the cleaved form significantly increased after photoactivation, suggesting the involvement of caspase 3 in the anti-survival effect induced by photoactivation of Ru(II) complexes in A2780 cells.

Finally, programed cell death and its involvement in the photoactivity of Ru(II) complexes was confirmed by measuring the cleavage of poly ADP-ribose polymerase (PARP), which is one of the major hallmarks of apoptosis (Figure 5). The specific inactivation of PARP by proteolytic cleavage was undetectable under dark conditions, while it was significantly appreciable after photoactivation of each Ru(II) complex, although to a different extent.

### 2.6. Mitochondrial Membrane Potential Is Lost after Photosensitization of Ru(II) Complexes

Mitochondria play a key role in the intrinsic pathway of apoptosis in mammalian cells, and mitochondrial membrane potential (Δψm) loss is considered an early event of the apoptotic process in some cellular systems [26]. For this reason, Δψm was analyzed after photosensitizer administration in A2780 cells using a cationic fluorescent probe, which accumulates in the negatively charged mitochondrial matrix, by laser-scanning confocal microscopy imaging. Figure 6 illustrates the distribution of functional mitochondria in A2780 cells in the presence of Ru(II) complexes under dark conditions or after photoactivation. The administration of photosensitizers under dark conditions did not affect Δψm or cell morphology. However, photoactivation of **Ru1**^2+^ and **Ru2**^2+^ caused a dramatic loss of Δψm in almost all the cells. Remarkably, this was accompanied by the appearance of a pyknotic morphology of the nuclei, possibly representing an initial phase of chromatin condensation prior to DNA fragmentation [27]. Nonetheless, A2780 cells treated with [Cu**Ru1**]^4+^ and [Cu_2_**Ru2**]^6+^ did not exhibit any mitochondrial change or decrease in Δψm in the presence or in the absence of photoactivation, thus ruling out the possible involvement of early loss of mitochondrial membrane potential in the pro-apoptotic effect of mixed Ru(II)-Cu(II) complexes. Given the cationic nature of Ru(II) complexes, mitochondrial localization subsequent to light-induced mitochondrial depolarization might occur. To analyze this possibility, we performed laser-scanning confocal microscopy employing a mitochondria-specific probe in A2780 cells incubated with **Ru1**^2+^, **Ru2**^2+^, [Cu**Ru1**]^4+^ or [Cu_2_**Ru2**]^6+^, followed by a colocalization test, to assess the possible localization of Ru(II) complexes into mitochondria. Confocal microscopy images showed a random distribution of Ru(II) complexes compared to mitochondria (Appendix A), thus ruling out the possibility that mitochondrial depolarization induced by photoactivation depends on Ru(II) complexes’ localization in these organelles.

### 2.7. Cytosolic ROS Production after Photosensitization of Ru(II) Complexes Ru1^2+^ and Ru2^2+^

With the purpose of dissecting the possible involvement of reactive oxygen species (ROS) production in the biological effect induced by **Ru1**^2+^ and **Ru2**^2+^ photosensitizers, confocal analysis was performed in A2780 cells employing the CM-H_2_DCFDA probe to detect cytosolic ROS after **Ru1**^2+^ and **Ru2**^2+^ administration upon photoactivation compared to the dark conditions. As shown in Figure 7, cytosolic ROS were almost completely undetectable after each **Ru1**^2+^ or **Ru2**^2+^administration in A2780 under dark conditions, even if Ru(II) complexes were efficiently internalized (red fluorescence). However, as early as 2 h after photoactivation, cytosolic ROS were detected in the majority of cell populations, notwithstanding at different degrees of intensity, thus suggesting a critical role of ROS in the proapoptotic action exerted by **Ru1**^2+^ and **Ru2**^2+^ photosensitizers.

## 3. Discussion

Increasing evidence reports that platinum-based anti-cancer drugs have severe side effects, such as myelotoxicity and peripheral neuropathy [28]. Moreover, ovarian cancer recurrence less than six months after the completion of platinum-based therapy is frequent, and prognosis is extremely poor [29]. The main reason for the dramatic failure of ovarian cancer treatments [30,31] relies on the intrinsic and acquired resistance to platinum-based chemotherapy. Therefore, efforts are needed in this research field to develop other potential anti-cancer drugs. Ru(II) polypyridyl complexes have shown remarkable anti-tumor activity coupled with advantages over platinum drugs, such as higher potency, lower toxicity, minor drug resistance, and they are expected to become a new generation of clinical metal anti-cancer drugs [11,32,33].

In our study, a series of RPCs-based photosensitizers for PDT application were investigated for the capability of inducing anti-tumor effects following photoactivation with low-energy light. Their biological potential was evaluated in the A2780 ovarian cell line, which was taken as a model of ovarian cancer [25]. Since PDT activity may result from different pathways, such as the production of ROS through *type I* and *II* mechanisms, four different Ru(II) polypyridyl complexes, namely **Ru1**^2+^, **Ru2**^2+^, [Cu**Ru1**]^4+^ and [Cu_2_**Ru2**]^6+^, featuring different modalities of activation, were evaluated in this study.

Among these metal complexes, **Ru1**^2+^ and **Ru2**^2+^ are highly luminescent and possess good (and comparable) abilities to sensitize the formation of singlet oxygen through *type II* reactions. On the other hand, the insertion of one or two Fenton-active Cu(II) ion/s in their corresponding heteronuclear Ru(II)-Cu(II) complexed species ([Cu**Ru1**]^4+^ and [Cu_2_**Ru2**]^6+^) results in less luminescent compounds, which can mainly promote the generation of perhydroxyl and hydroxyl radicals via *type I* reactions. The coordination of Cu(II) ion/s in these latter two compounds is remarkably strong (LogK values between 15.34 and 27.6), making any transmetalation process unfavorable in conditions mimicking the cellular environment. Importantly, the peculiar polyamine frameworks confer to all these compounds excellent solubilities in water, a key requisite for their application in the biomedical field.

In a previous report, the Ru(II)-arene complex [Ru(η6-p-cymene)Cl_2_] (RAPTA-C) was tested for efficacy in combination with the epidermal growth factor receptor inhibitor erlotinib, demonstrating an efficient anti-angiogenic and anti-tumor activity [34]. The therapeutic potential of these compounds and their combination was further confirmed in preclinical in vivo models of chicken chorioallantoic membrane grafted with A2780 tumors and in mice bearing A2780 tumors, highlighting the tumor growth inhibition and anti-angiogenic effect [34].

In line with the literature data, the RPCs investigated herein were found to possess negligible cytotoxicity without light irradiation in non-cancer and cancer cells, respectively, converted in turn in a tight dose-dependent and significant anti-tumor action upon photoactivation. In particular, a marked increase in the activity was observed upon light activation of **Ru1**^2+^ and **Ru2**^2+^, in agreement with the good singlet oxygen sensitizing properties of these compounds. Among the mixed heteronuclear complexes, [Cu**Ru1**]^4+^ displayed the sharpest photoactivity. Interestingly, this suggests that, beyond the singlet oxygen sensitization, alternative oxidative pathways must be accessible to heteronuclear compounds, leading to similar photoinduced effectiveness compared to the one of copper-free complexes. However, further efforts will be needed to obtain further insights into these processes, which likely occur under biological conditions and appear to be hard to mimic in cell-free experiments. Moreover, the significant differences in activity observed between [Cu**Ru1**]^4+^ and [Cu_2_**Ru2**]^6+^ suggest that other less predictable features, such as differences in the chemical structures, may play a critical role in the biological behavior of such compounds.

The investigated RPCs also displayed good capacities to be internalized by A2780 cancer compared to non-cancer cells, with **Ru1**^2+^ and **Ru2**^2+^ being the most effective. This result is of great interest, especially considering that similar RPCs-based PSs were shown to poorly penetrate the cell membrane, and additional expedients, such as ion pairing with suitable lipophilic counter-anions, were necessary to augment the cellular uptake [35]. In the present study, it is reasonable to assume that the significant cellular uptake by cancer cells is associated with the presence of the polyamino macrocyclic frameworks **L’** and **L’’** of the Ru(II) compounds, which may impart optimal chemical–physical features for cellular internalization, such as hydrophilicity and total charge of the compounds. Even if the precise mechanism by which the Ru(II) compounds are internalized in A2780 cells was not investigated in detail, their uptake appears to be independent of the Rab5-dependent early endosome pathway. Nonetheless, taking into consideration the hydrophilic chemical nature of RPCs and their observed localized cellular distribution, the occurrence of an efficacious passive transport can be excluded, rather pointing at alternative endocytotic events accounting for RPC uptake. Indeed, this is in line with the literature data showing the occurrence of specific cellular transport of cytotoxic metallodrugs [36].

Apoptosis is clearly advantageous for the organism, since, during apoptosis, the cell membrane remains intact, thus preventing the release of intracellular content. Hence, the elucidation of the mechanisms that triggered cell death after the light activation of Ru(II) complexes appears to be crucial. In A2780 cells, the photoactivation of all the tested RPCs caused a potent caspase 3 activation, as well as both caspase 3 and PARP cleavage, while the RPCs were ineffective in the absence of photosensitization, pointing at a crucial role for programed cell death in the anti-tumor activity of these systems.

Mitochondria might be seen as a gatekeeper to entrap pro-apoptotic proteins and prevent the release and activation of these proteins in the cytosol [37]. In particular, the exit of pro-apoptotic proteins from the mitochondria activates caspase proteases. Of note, the opening of the mitochondrial permeability transition pore has been demonstrated to induce Δψm depolarization, loss of oxidative phosphorylation and release of apoptogenic factors. Thus, a distinctive feature of apoptosis can be represented by the disruption of the normal mitochondrial function, especially changes that affect the Δψm. In some apoptotic systems, the loss of Δψm may be an early event in the apoptotic process [38]. Here, using confocal microscopy, it was demonstrated that **Ru1**^2+^ and **Ru2**^2+^, but not [Cu**Ru1**]^4+^ or [Cu_2_**Ru2**]^6+^, caused a dramatic loss of Δψm depending on photoactivation and that this effect was accompanied by ROS production in the cytosol as soon as 2 h after light irradiation, thus upstream to the pro-apoptotic stimuli.

Lastly, although previous in vitro studies underlined the capacity of RPCs to effectively interact and damage plasmid DNA upon irradiation [16], in this work, we did not detect RPCs in the nucleus, at least within 24 h of incubation, since these compounds were found distributed into segregated areas of the cytosol. This finding suggests that the induced programed cell death is likely independent of the PSs–DNA interaction.

In summary, the present study demonstrates that all four synthesized Ru(II) complexes are effectively internalized into the ovarian cancer A2780 cells, and their administration, regardless of the low dark cytotoxicity, induces a specific photoactivation-dependent cell death, with the extent of cytotoxicity that varies slightly depending on the chemical structures of RPCs. Apoptosis emerged as the main mechanism of light-mediated cellular death. In particular, among the four compounds, **Ru1**^2+^ and **Ru2**^2+^ profoundly altered mitochondrial activity after photoactivation, accompanied by cytosolic ROS production.

## 4. Materials and Methods

### 4.1. Materials

All materials used for the preparation of ruthenium compounds were of reagent grade and used as received, unless otherwise specified.

### 4.2. Synthesis of Ru(II)-Complexes

Ruthenium compounds **Ru1**^2+^, **Ru2**^2+^, [Cu**Ru1**]^4+^ and [Cu_2_**Ru2**]^6+^ were synthesized according to the procedures previously described [16,39]. Briefly, **Ru1**^2+^ and **Ru2**^2+^ were prepared by direct reaction of the intermediate (phen)_2_**Ru**Cl_2_ with the bidentate **L’** o **L’’** ligands, in ethylene glycol and under microwave irradiation. The resulting complexes were then dissolved in concentrated HCl and precipitated as their respective hydrochloride salts [**Ru1**]Cl_2_·5HCl·2H_2_O and [**Ru2**]Cl_2_·6HCl·2H_2_O, following the addition of ethanol.

The mixed Ru(II)/Cu(II) complexes were obtained as the perchlorate salts [Cu**Ru1**](ClO_4_)_4_·4H_2_O and [Cu_2_**Ru2**](ClO_4_)6·3H_2_O, by adding equimolar amounts of Cu(ClO_4_)_2_ to aqueous solutions of **Ru1**^2+^ or **Ru2**^2+^ at pH 6.5 and following the slow evaporation of the solvent at r.t.

### 4.3. Potentiometric Measurements

The acid-base behavior and the binding ability toward Cu^2+^, Zn^2+^, Ca^2+^, Na^+^ and Mg^2+^ of ruthenium complexes were investigated by means of potentiometric measurements in NMe_4_Cl 0.1 M at 298 ± 0.1 K by using the equipment and methods previously described [40,41,42,43].

### 4.4. EPR Measurements

X-band electron paramagnetic resonance (EPR) experiments were performed by using a Bruker Elexsys E500 spectrometer. All the spectra were acquired at room temperature by using the same modulation frequency (100 kHz), modulation amplitude (1 G), microwave power (~0.2 mW, 30 dB) and receiver gain (60 dB). The magnetic field was calibrated with a crystal of DPPH.

### 4.5. Cell Culture

Human A2780 ovarian cancer cell culture (ECACC 93112519) was maintained in RPMI-1640 medium containing 10% fetal bovine serum (FBS), 100 μg/mL streptomycin, 100 U/mL penicillin and 2 mM L-glutamine, at 37 °C in 5% CO_2_, as previously reported [44]. All cell culture reagents were purchased from Merck Life Science (Darmstadt, Germany), including phosphate-buffered saline (PBS). A2780 cells were shifted to RPMI without serum supplemented with 1 mg/mL Bovine Serum Albumin (BSA) and treated with each Ru(II) complex (0.1, 1 and 10 µM) for 24 h. After incubation, cells were photoactivated with a 30 W three-arm LED light lamp (430–470 nm emission, 30 W) for 20 min at the distance of 5 cm from the cell culture plate and then kept in the incubator at 37 °C, 5% CO_2_. Cells were washed twice with PBS and then collected after photoactivation at different times, which depended on each kind of experiment.

### 4.6. ICP-AES Measurements

A Varian 720-ES axial Inductively Coupled Plasma Atomic Emission Spectrometer (ICP-AES) was used to determine the Ru contents in the samples. Measurements were performed in triplicate, and each sample was spiked with 1.0 ppm of Ge, used as an internal standard. The calibration standards were prepared by gravimetric serial dilution from commercial stock standard solutions of Ru at 1000 mg L^−1^ (Honeywell Fluka). For Ru determination, the 267.876 and 245.657 nm wavelengths were used, whereas the line at 209.426 nm was considered for Ge. The operating conditions were optimized to obtain maximum signal intensity, and between each sample, a rinse solution containing 2% *v*/*v* of HNO_3_ was used.

### 4.7. MTT Reduction Assay for Cell Survival

A2780 viability was evaluated by the MTT method, as previously described [45].

### 4.8. Caspase-3 Activity Assay

A2780 cells were seeded in 6-well plates (100.000 cells/well) and after 24 h were incubated with 10 µM of each Ru(II) complex in serum-deprived culture media, then light-irradiated for 20 min, as described above. After 24 h of photoactivation, cells were washed twice with PBS, collected and analyzed, as previously described [46].

### 4.9. Western Blot Analysis

A2780 lysates were quantified for total protein content by the Bradford Protein assay, resuspended in Leammli’s sodium dodecyl sulphate (SDS) sample buffer, and subjected to SDS-PAGE and transferred to PVDF membranes, as previously described [46].

### 4.10. Laser-Scanning Confocal Microscopy

A2780 cells were seeded on microscope slides and treated with each Ru(II) complex (10 µM). To evaluate the internalization of Ru(II) complexes, cells were incubated at three different times, 15 min, 6 h and 24 h, washed with PBS and fixed in 2% paraformaldehyde in PBS for 20 min. Ru(II) complexes excitation was performed using 405 nm laser diode, acquiring emission in the range of 600/620 nm.

MitoTracker Red CMXRos (#M7512; Ex/Em: 579/599 nm) and CM-H_2_DCFDA (#C6827; Ex/Em: ∼492–495/517–527 nm) probes (Invitrogen, Thermo Fisher Scientific INC, Waltham, MA, USA) were used to detect the mitochondrial membrane potential and ROS production, respectively. After 24 h incubation of Ru(II) complexes at 37 °C, cell slides were photoactivated or not for 20 min and incubated for 2 h at 37 °C, 5% CO_2_ in humidified atmosphere. Probes were diluted in RPMI medium without phenol red, incubated for 30 min at 37 °C in dark and then fixed in 2% paraformaldehyde, as suggested by the manufacturer’s instruction. After 30 min at room temperature, slides were incubated with a permeabilization and quenching solution, obtained by adding Triton 0.1% X-100 and ethanolamine (1:165) in PBS. The DAPI solution was administered to cell slides to detect the nuclei. Slides were mounted by using the Fluoromount Aqueous Mounting Medium (Sigma-Aldrich, Saint Louis, MA, USA), and images were obtained using a Leica SP8 laser-scanning confocal microscope (Leica Microsystems GmbH) using a 63x oil immersion objective.

### 4.11. Statistical Analysis

Densitometric analysis of Western blot bands was performed using the ImageJ software, and graphical representations were obtained by GraphPad Prism 5.0 (GraphPad Software, San Diego, CA, USA). Statistical analysis was performed using one-way or two-way ANOVA analysis of variance followed by the Bonferroni post hoc test. Asterisks indicate statistical significance.

## 5. Conclusions

In conclusion, our study identifies Ru(II)-polypyridyl complexes as challenging tools to be further investigated in the research of new therapeutic strategies to overcome chemoresistance in epithelial ovarian cancers and provides further insights on the biological behavior of these complexes, which rely on ROS production and altered mitochondrial function to trigger pro-apoptotic effects.

## Data Availability

All the data are reported in the manuscript.

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
