# Peer review of "Highly Charged Ru(II) Polypyridyl Complexes as Photosensitizer Agents in Photodynamic Therapy of Epithelial Ovarian Cancer Cells"

_ijms, 2022, doi:10.3390/ijms232113302_

Round 1
Reviewer 1 Report
The article “Highly charged Ru(II) polypyridyl complexes as photosensitizer agents in photodynamic therapy of epithelial ovarian cancer cells” by Conti L. et al. is an experimental work that investigates the possible anticancer effects of a series of new highly charged Ru(II)-polypyridyl complexes as alternative photosensitizers in photodynamic therapy (PDT), capable of forming singlet oxygen upon irradiation and producing reactive oxygen species (ROS) with cytotoxic and antitumor effects in human ovarian cancer A2780 cells, studies performed both in the absence and respectively, in the presence of photoirradiation. It was found in the case of Ru12+ and Ru22+ the loss of Δψm shortly after photoactivation and the production of ROS, thus supporting the initiation of apoptosis through type II photochemical reactions. The authors tried to prove that photosensitizers based on Ru(II)-polypyridyl complexes represent challenging tools to be further investigated in identifying new therapeutic approaches to overcome chemoresistance to platinum derivatives in some epithelial ovarian cancers and to discover new drugs for the treatment of the recurrent ovarian cancer.
However, the article has several shortcomings as follows:
1. The title is not correctly written in the all-caps title style.
2. For the keywords: The authors did not adhere to MeSH for choosing keywords and have several imperfections in keyword formulations. The authors also included some keywords that are already in the TITLE and should not have been included.
3. A schematic representation of the installation and the way of irradiating the cells is necessary for the reader.
4. More comprehensive arguments and explanations on the presented results in all the Figures would be very welcome.
5. The manuscript is not overloaded with unnecessary information, but it could be improved in the Discussion.
6. More comparative arguments regarding the actual results obtained are missing!
7. Please, rephrase lines 573-575: "Even if the precise mechanism by which the Ru(II) compounds are internalized in A2780 cells has not been investigated in detail, we are confident that the PS uptake does not rely on the mechanism of passive diffusion"!!!
8. Superficial formulations of the type "our study suggests a potential possibility for Ru(II) complexes to be employed" do not correspond to a serious and honest scientific work!
Please, see lines: 754-755, and express the idea in an alternative, better way.
9. The editing was done in haste and with mistakes, for example:
A. In the following lines, the pause before the cited reference is missing, please correct in the following lines:
For example, incorrect: “death in reproductive women[1,2].”
Please, correct as:
“death in reproductive women [1,2].” - a space before the square bracket is absolutely necessary!
Please, correct the same mistakes in the lines: 37, 39, 43, 47, 50, 57, 75, 96, 180, 204, 212, 234, 253, 283, 509, 511, 512, 517, 521, 546, 569, 577, 596, 604, 627, 641, 644, 650, 702, etc.
B. There are subchapter titles written with a "period" at the end:
Please, correct as follows:
2.1 – line 122
2.2 – line 200
2.4 – line 343
2.5 – line 394
2.6 – line 440
4.6 – line 669
10. All references must be double-checked.
11. A list of abbreviations must be completed and reviewed carefully and may be better presented in a table format in the end.
In general, I commend the authors for the preparation of this article, which required a lot of experimental work, but a minor revision is needed!
More English and style issues could be improved by a native English speaker at a final reading on your MDPI platform.
Reviewer 2 Report
The Article is dedicated to urgent problem of treatment of ovarian cancer associated with resistance to platinum- and taxane-based chemotherapies. The Authors explored the potential anti-cancer effect of a series of highly charged Ru(II)-polypyridyl complexes as photosensitizers in photodynamic therapy (PDT). Ru12+ and Ru22+, as well as their corresponding dinuclear metal complexes with the Fenton active Cu(II) ion/s ([CuRu1]4+ and [Cu2Ru2]6+ were investigated for protonation, metal binding and stability at physiological pH value and Reactive Oxygen Species (ROS) Production. A2780 ovarian cancer cells as well as a non-cancer cell line, namely C2C12 myoblasts were studied for internalization of Ru(II) complexes, and laser-scanning confocal microscopy was performed by exploiting the intrinsic fluorescence properties of Ru(II) compounds. Their cytotoxic and antitumor effects were evaluated on human ovarian cancer A2780 cells both in the absence or presence of photoirradiation, respectively. All the compounds tested were well tolerated under dark condition whereas they switched to exert
antitumor activity following photoirradiation. The specific effect was mediated by the onset of programmed cell death but, only in case of Ru12+ and Ru22+, it was preceded by the
loss of Δψm soon after photoactivation and ROS production, thus supporting the occurrence of apoptosis via type II photochemical reactions.
Thus Ru(II)-polypyridyl-based photosensitizers represent challenging tools to be
further investigated in the identification of new therapeutic approaches to overcome innate chemoresistance to platinum derivatives of some ovarian epithelial cancer and to
find innovative drugs for recurrent ovarian cancer.
The Introduction, as a whole Article, is written briefly and well understandable. The background and the goal are clear. Materials and methods were chosen adequately and well described. All Tables and Figures are informative and designed appropriately. The Conclusions correspond to the results obtained and Discussion is interesting to read.
The only small misprint is at line 76: L’ = 4,4’-bis-[methylene-…. May be L’’ ?
The article may be accepted for publication.

Reviewer 3 Report
The manuscript explored the potential anti-cancer effect of a series of highly charged Ru(II)-polypyridyl complexes as photosensitizers in photodynamic therapy (PDT) that are able to efficiently sensitize the formation of singlet oxygen upon irradiation and to produce reactive oxygen species (ROS) in their corresponding dinuclear metal complexes.
Their cytotoxic and antitumor effects were evaluated on human ovarian cancer A2780 cells both in the absence or presence of photoirradiation, respectively. All the compounds tested were well tolerated under dark condition whereas they switched to exert antitumor activity following photoirradiation. I have the following comments that should be addressed before the publication:
a) The manuscript is an extension of the article previously published by the same authors (Chem.Eur.J.2019,25,10606–10615) (reference 17) on a different cancer cell line.
b) These molecules could also act as intercalating agents on DNA. Can the authors speculate an interaction with DNA as mechanism of action?
c) Based on the antiproliferative data reported on A2780 human ovarian cancer cells, for the two [CuRu] complexes the Fenton-active copper center plays a synergetic role with light activation in the development of cytotoxic ROS species, providing additional mechanisms for the oxidative damage to biological targets.
d) After incubation, the cells were washed thoroughly to remove the excess of extracellular compounds?
e) Is it possible to study the intracellular localization of the Ru derivatives by means of fluorescence microscopy?
Round 2
Reviewer 3 Report
The authors have answered to all my questions. The manuscript deserves to be published in the present form.